# D2I and F9Y Mutations in the NS1 Protein of Influenza A Virus Affect Viral Replication via Regulating Host Innate Immune Responses

**DOI:** 10.3390/v14061206

**Published:** 2022-06-01

**Authors:** Mengqi Yu, Yanna Guo, Lingcai Zhao, Yuanlu Lu, Qingzheng Liu, Yinjing Li, Lulu Deng, Zhiyu Shi, Haifeng Wang, Samar Dankar, Jihui Ping

**Affiliations:** 1MOE Joint International Research Laboratory of Animal Health and Food Safety, Engineering Laboratory of Animal Immunity of Jiangsu Province, College of Veterinary Medicine, Nanjing Agricultural University, Nanjing 210095, China; ymq13515127867@163.com (M.Y.); gyn13655183568@163.com (Y.G.); 2018207020@njau.edu.cn (L.Z.); luyuanlu1212@163.com (Y.L.); vetliuqingzheng@126.com (Q.L.); leeyinjing@163.com (Y.L.); dll980620@163.com (L.D.); shizhiyu@njau.edu.cn (Z.S.); 2Changjia (Shanghai) Agricultural Science and Technology Co., Ltd., Shanghai 201414, China; laker88wang@163.com; 3Department of Biochemistry, Microbiology and Immunology, Faculty of Medicine, University of Ottawa, Ottawa, ON K1V 8M5, Canada; samardankar@gmail.com

**Keywords:** influenza A virus, NS1 protein, pathogenicity, IFN

## Abstract

Influenza A viruses (IAV) modulate host antiviral responses to promote viral growth and pathogenicity. The non-structural (NS1) protein of influenza A virus has played an indispensable role in the inhibition of host immune responses, especially in limiting interferon (IFN) production. In this study, random site mutations were introduced into the NS1 gene of A/WSN/1933 (WSN, H1N1) via an error prone PCR to construct a random mutant plasmid library. The NS1 random mutant virus library was generated by reverse genetics. To screen out the unidentified NS1 functional mutants, the library viruses were lung-to-lung passaged in mice and individual plaques were picked from the fourth passage in mice lungs. Sanger sequencing revealed that eight different kinds of mutations in the NS1 gene were obtained from the passaged library virus. We found that the NS1 F9Y mutation significantly enhanced viral growth in vitro (MDCK and A549 cells) and in vivo (BALB/c mice) as well as increased virulence in mice. The NS1 D2I mutation attenuated the viral replication and pathogenicity in both in vitro and in vivo models. Further studies demonstrated that the NS1 F9Y mutant virus exhibited systematic and selective inhibition of cytokine responses as well as inhibited the expression of IFN. In addition, the expression levels of innate immunity-related cytokines were significantly up-regulated after the rNS1 D2I virus infected A549 cells. Collectively, our results revealed that the two mutations in the N-terminal of the NS1 protein could alter the viral properties of IAV and provide additional evidence that the NS1 protein is a critical virulence factor. The two characterized NS1 mutations may serve as potential targets for antiviral drugs as well as attenuated vaccine development.

## 1. Introduction

Influenza viruses belong to the family Orthomyxoviridae; there are four main types (influenza A, B, C and D) [1]. Influenza A virus is an important human and animal pathogen involved in global pandemics and annual epidemics with huge economic losses and public health concerns [2]. Every year, approximately 290,000–650,000 people die from respiratory diseases caused by influenza [3]. Influenza A viruses harbor a genome comprising eight single-stranded, negative-sense RNA segments that each encodes one or two viral proteins [4]. Based on the antigenicity of the hemagglutinin (HA) and neuraminidase (NA) surface proteins, IAVs can be divided into 18 different HA subtypes (H1–H18) and 11 different NA subtypes (N1-N11) [5]. The IAV non-structural protein 1 (NS1) is expressed from the full-length transcript of gene segment 8 that overlaps with the spliced nuclear export protein (NS2/NEP) transcript [6]. The length of the NS1 protein varies from 215–237 amino acids, depending on the virus strain [7]. The NS1 protein is divided into two functional domains: the N-terminal RNA binding domain (RBD) and the C-terminal effector domain (ED), which interacts with a variety of host factors [8,9]. The RBD rotates to form a six-helix homodimer that binds to double-stranded RNA (dsRNA). The ED of most NS1 proteins contains nuclear output signals [10].

NS1 is a multifunctional protein that interacts with multiple cellular factors and antagonizes the host antiviral response during viral infection [9]. The major role of the NS1 protein is the inhibition of both interferon (IFN) production and IFN-stimulated proteins’ signaling via multiple mechanisms. Briefly, the NS1 protein interacts with viral dsRNA, therefore, preventing the activation of pattern recognition receptors (PRRs), including RIG-I, hence, limiting IFN transcription. In parallel, sequestering the viral dsRNA also prevents the activation of the dsRNA-dependent protein kinase R (PKR), and 2′5′-oligoadenylate synthetase (OAS)/RNase L [11,12,13], shown to be a key initiator of the type I interferon response in infected epithelial cells, and therefore, limits the activation of IFN transcription [14,15]. NS1 can also interact directly with RIG-I in the absence of RNA binding and, therefore, inhibits the conformational change of RIG-I required for MAVS activation [16]. Moreover, NS1 also inhibits IFN production post-transcriptionally, by limiting the 3′-end processing of the host mRNA, including IFN mRNA, by binding to cleavage and polyadenylation specificity factor 30 (CPSF30) [17] and poly-A-binding protein II (PABPII) [18]. Both CPSF30 and PABPII factors are required for host mRNA processing, resulting in the accumulation of IFN pre-mRNAs in the nucleus of infected cells [19]. In addition, NS1 also antagonizes the IFN signaling response by regulating other host factors, such as phosphoinositide 3-kinase (PI3K) activity, Crk-like protein (CRKL), and the JAK-STAT signaling pathway [20,21,22,23,24]. The NS1 is also involved in enhancing viral protein synthesis by interacting with the viral mRNA [25], the translation initiation factors eIF4GI (eukaryotic initiation factor 4GI) [26], as well as the PABPI (poly-A-binding protein 1) [27]. Therefore, NS1 is an important influenza virus protein for the suppression and evasion of innate immunity; changes in its amino acid sites have had an impact on its biological properties, with some site-specific mutations significantly enhancing the pathogenicity of influenza A viruses [28].

Here, we focus on screening the NS1 unidentified mutations and illustrate their effects on IAV virulence and pathogenicity, as well as the molecular mechanisms exploited by NS1 to antagonize the host innate immunity, providing potential targets and molecular mechanisms for the development of new antiviral drugs. A random mutagenesis technique was used to construct an NS1 plasmid library and then generate a virus library in an A/WSN/1933(WSN, H1N1) backbone. After four rounds of lung-to-lung passages in mice, individual viral clones were purified and sequenced. A bunch of NS1 mutations were confirmed and deep analyzed. We found NS1 random mutations selected upon mouse adaptation to be multifunctional, where NS1 mutation F9Y increased virulence in the mice model, replicative efficiency in vivo and in vitro, as well as enhanced properties of IFN antagonism. However, mutation D2I in the NS1 protein attenuated the replication and virulence of the WSN virus in vitro and in vivo; IFN-β antagonist assays indicated that the D2I mutation also attenuated the IFN antagonistic ability compared to the NS1-WT protein.

## 2. Materials and Methods

### 2.1. Virus and Cells

Human embryonic kidney (HEK293T) cells were cultured in Dulbecco’s Modified Eagle’s medium (DMEM; Gibco, Invitrogen, Carlsbad, CA, USA) containing 10% fetal bovine serum (FBS; Gibco), 0.2% NaHCO_3_, 100 µg/mL streptomycin, and 100 IU/mL penicillin (Gibco) at 37 °C with 5% CO_2_. Madin–Darby Canine Kidney (MDCK) cells were supplemented with DMEM containing 10% new born calf serum (FBS; Sigma, Ronkonkoma, NY, USA). A549 cells were grown in DMEM containing 10% new born calf serum (FBS; Sigma). A/WSN/1933(H1N1, WSN) and its NS1 mutant viruses were generated with a normal NS segment by reverse genetics and propagated in MDCK cells and stored in aliquots at −80 °C. The rescued viruses were confirmed by Sanger sequencing, and viral titers were determined by plaque assay in MDCK cells.

### 2.2. Plasmid Construction

The gene fragment including WSN NS1 ORF followed by the porcine teschovirus 1 (PTV-1) 2A autoproteolytic cleavage site (A T N F S L L K Q A G D V E E N P G P) and linked with the entire NEP ORF of WSN were cloned into the RNA polymerase I vector pHH21, namely pHH21-WSN-NS_linear_. The NEP gene segment of WSN was fused to the C-terminal of the ORF of enhanced green fluorescent protein (eGFP) via overlap PCR and then cloned into the protein expressing vector pCAGGS to generate NEP’s fluorescent expression plasmid.

### 2.3. Generation of NS1 Random Mutated Plasmid Library

The ORF for the NS1 of pHH21-WSN-NSlinear were randomly mutagenized by error-prone PCR using a GeneMorph II Random Mutagenesis Kit (Agilent). The PCR reaction conditions (including primers amounts, PCR cycles, Mg^2+^ concentration and the target DNA template amounts) were optimized to generate 1–3 amino acid changes in each NS1 ORF molecule. The wild type NS1 segment of pHH21-WSN-NS_linear_ plasmid was replaced with randomly mutated NS1. PCR products generated by PCR based cloning technique were used to construct a NS1 random mutant plasmid library.

### 2.4. Site-Directed Mutagenesis

Point mutant plasmids were generated by using a site-directed mutagenesis kit (Beyotime, Shanghai, China). A total of 50 ng plasmid DNA was used for the reaction with the mutagenic oligonucleotides. All point mutations were verified via Sanger sequencing.

### 2.5. Virus Rescue and Virus Library Generation

All of the viruses and the NS1 library viruses were generated using a “twelve plasmids” reverse genetics system [29]. Briefly, 1 μg of each protein expression plasmid (pCAGGS-WSN-PB2, pCAGGS-WSN-PB1, pCAGGS-WSN-PA and pCAGGS-WSN-NP) and 0.2 μg of each viral RNA transcription plasmid (pHH21-WSN-PB2, pHH21-WSN-PB1, pHH21-WSN-PA, pHH21-WSN-HA, pHH21-WSN-NP, pHH21-WSN-NA, pHH21-WSN-M and pHH21-WSN-NS (or pHH21-WSN-NS_linear_, pHH21-WSN-NS1 mutants, or NS1 random mutant library plasmid) were combined with 12 μL Lipofectamine 2000 (2 μL per μg DNA, Invitrogen) according to the manufacturer’s instructions and incubated for 30 min at room temperature. The plasmids and the transfection reagent mixtures were added into a 70% confluent monolayer of 293T cells; 16 h post transfection, the transfection mix was replaced by fresh opti-MEM (Invitrogen, Carlsbad, CA, USA) supplemented with TPCK-trypsin (1 μg/mL). Forty-eight hours post-transfection, the cell supernatant was collected and inoculated into MDCK cells for virus propagation at 37 °C for 48 h. The clarified supernatants were collected, and the viral titers were assessed by a plaque assay in MDCK cells. The NS genes of the rescued viruses were verified by sequencing analysis.

### 2.6. Plaque Assay

The infectious titers of the recombinant viruses were determined by plaque assay according to the following described procedures. Briefly, the viruses were serially 10-fold diluted in 1×CDMEM and inoculated onto MDCK cell monolayers. Twelve well plates of confluent monolayers of MDCK cells were washed twice with 1xPBS and then infected with 100 μL of the different virus dilutions in duplicate for each dilution. After incubation at 37 °C for 1 h, the cells were overlaid with 1×CDMEM containing 1.8% Sea Plaque agarose (supplemented with 1 μg/mL TPCK-trypsin) and incubated at 37 °C. Plaque formation was observed two days post-infection (dpi).

### 2.7. Viral Growth Kinetics in Cell Culture

Confluent MDCK or A549 cells were infected with the indicated viruses at MOI = 0.001 or 0.1, respectively, for 1 h. The cells were then washed twice with 1xPBS and then incubated in DMEM containing 0.3% BSA, 20 mM Hepes and 0.5–1 µg/mL of TPCK-trypsin. Culture supernatants were collected at different time points and the infectivity of the progeny viruses was determined by plaque assay in MDCK cells.

### 2.8. Mouse Experiments

Six to eight-week-old female BALB/c mice were purchased from Xipuer-bikai Experimental Animal Company (Shanghai, China). Five mice per group were anesthetized with isoflurane and intranasally inoculated with 5 × 10^3^ or 10^4^ PFU of WSN or rNS1 mutant viruses in a volume of 50 μL. PBS-infected mice were used as a negative control group. Survival and body weight loss were monitored daily for 14 days. Mice were euthanized when body weight loss reached 25% of the baseline weight (the weight on the day of infection). To assess the viral replicative efficiency, six mice from each group were intranasally inoculated with 10^4^ PFU of the indicated viruses. Three mice from each were euthanized on 3 and 5 dpi, the ratio of lung to the whole mouse body weight was calculated and then the lungs were suspended in 1 mL 1xPBS and subsequently homogenized. The virus titers were evaluated by plaque assay. For the pathological examination, the BALB/c mice were intranasally infected with 10^4^ PFU of the indicated viruses, and the lungs were collected at 3 dpi and fixed in 10% phosphate-buffered formalin. Subsequently, the lungs were embedded in paraffin, sectioned at a thickness of 5 μm, stained with hematoxylin and eosin (H & E), and examined under light microscopy (Nikon, Tokyo, Japan) for histopathologic changes.

### 2.9. Luciferase Reporter Assay

The reporter plasmids expressing firefly luciferase under the control of the IFN-β promoter were used to measure the relative ability of the NS1 proteins to suppress the transfection-induced IFN-β promoted luciferase. A second reporter plasmid expressing the constitutive Renilla luciferase reporter (Promega, Madison, WI, USA) was used to measure the general host expression in every transfection experiment. Briefly, a 70% confluent monolayer of HEK 293T cells in 24-well plates were co-transfected with 0.2 μg/well of reporter plasmid IFN-β-luc, 0.2 μg/well of plasmid pCAGGS-WSN-NS1 or pCAGGS-WSN-NS1 mutant (or empty pCAGGS vector as a negative control) plasmids and 10 ng/well pRL-TK plasmid. Twenty-four hours following transfection, the cells were infected with Sendai virus. Cells were lysed at 12 h post-infection (hpi), and the firefly luciferase and Renilla luciferase activities were determined with the Dual-Luciferase reporter assay kit (Promega) according to the manufacturer’s protocol. Data were presented as relative firefly luciferase activities normalized to Renilla luciferase activities and were representative of three independent experiments.

### 2.10. qRT-PCR

Total RNA was extracted from the viral infected A549 cells or mouse lung homogenate using Trizol reagent (Invitrogen) according to the manufacturer’s instructions. The relative expression levels of viral mRNA and vRNA in virus-infected cells were detected by qRT-PCR. Cytokines and chemokines IL-4, IL6, IL-4, IFN-β, TNF-α, MIP1α and MCP-1 in the lungs of the infected mice were tested at 3 dpi by qPCR. The corresponding primers are available upon request. Two micrograms of total RNAs were reverse transcribed with oligo (dT). The cDNA was diluted with ddH_2_O, and the qRT-PCR was performed on a LightCycler^®^ 96 system in 20 μL reaction mixture containing 2 μL of cDNA, 4 μM of each primer, and 10 μL of 2×PCR master Mix. The relative expression of mRNAs was normalized to that of the GAPDH levels. The qPCR reaction was conducted with the following program: 1 cycle at 95 °C for 5 min, followed by 40 cycles of 95 °C for 10 s and 60 °C for 30 s. The relative mRNA levels of the indicated genes were calculated by the 2^−ΔΔCt^ method. Each experiment contained three technical replicates per sample, and three experimental replicates were performed.

### 2.11. Immunofluorescence

HEK293T cells were seeded on glass coverslips and transfected with the indicated plasmids. Following incubation, the cells were fixed with 4% paraformaldehyde and permeabilized with a solution of 1xPBS, containing 0.2% Triton X-100 for 10 min (Sigma). Then, the cells were stained with the 4,6-diamidino-2-phenylindole (DAPI) for 10 min. Finally, images were acquired using confocal microscopy (Nikon, Tokyo, Japan) equipped with a micro-objective (Plan Apo 60×/1.40, oil immersion, Nikon, Tokyo, Japan) and microscope eyepiece (CFI, 10×/22, Nikon).

### 2.12. Statistical Analysis

The data were expressed as the means ± standard deviations (SD). The significance was determined with the two-tailed independent Student’s *t*-test. A *p* value < 0.05 was considered statistically significant. Quantification of fluorescence signal was performed in ImageJ by counting 20–30 cells for each construct, and the ratio of IntDEN (Integrated density) within nuclear to cytoplasmic indicates nuclear export activity of NS2 protein.

### 2.13. Ethics Statement

The mice were handled humanely according to the rules described by the Animal Ethics Procedures and Guidelines of the People’s Republic of China and the Institutional Animal Care and Use Committee of Nanjing Agricultural University [SYXK(Su) 2017-0007].

## 3. Results

### 3.1. Virus Library Screening for Unidentified Functional NS1 Mutations

Although a number of NS1 mutations have been identified that antagonize the host immune response and promote viral replication in the host cells, it is not known whether there exists additional mutations carrying out similar or novel functions to improve replication and virulence. Therefore, in this study, we used a high-throughput mutagenesis strategy to screen for unidentified NS1 mutations associated with high replicative efficiency and virulent properties. Specifically, a linear NS gene segment of A/WSN/1933 (WSN, H1N1) was constructed first to avoid introducing unwanted mutations into the NEP gene (Figure 1A). NS1 random mutated cDNAs were generated by error-prone PCR and then inserted back into the pHH21-WSN-NS_linear_ vector to construct an NS1 library plasmid. Through sequencing of the individual plasmid from the library confirmed that the colony included 1–3 amino acid substitutions in the NS1 ORF. In addition, the NS1 library contained 1.42 × 10^6^ colony-forming-units. The random mutated virus library was then generated by reverse genetics with the NS1 library plasmid in the WSN backbone. The viral titer of the 293T transfected supernatants confirmed that the diversity of the NS1 library virus was 2.65 × 10^5^ PFU. To screen the highly replicative and virulent variants from the library virus, the library viruses were lung-to-lung passaged in BALB/c mice for 4 rounds. The body weight loss of each passage was monitored for 3 days (Figure 1B). The mice in the fourth-round passage showed 17.4% body weight loss. Therefore, the mice in this group were sacrificed and the infected lungs were collected and homogenized. Subsequently, two rounds of plaque assay were performed to isolate the individual viral colony, and 24 purified plaques were selected for further sanger sequencing. The sequencing results in respect to the NS1 gene showed that 13 mutant viruses were identified, of which there were 8 different combinations of single, double, or triple mutations within the NS1 gene (Figure 1C). Interestingly, among the 13 mutant viruses, the most common mutant was Phenylalanine (F) to Tyrosine (Y) at position 9 of NS1, see Appendix A.

To analyze whether these identified NS1 mutations exist elsewhere in H1N1 IAVs in nature, the NS1 genes of 2304 human, 231 avian and 2269 swine origin strains were downloaded from the Influenza Research Database (https://www.ffludb.org/ (accessed on 15 December 2021)) and analyzed. The NS1 sequence analysis revealed that these screened NS1 mutations exist in H1N1 IAVs, although they are not the most prevalent amino acid substitution. The percentage of the dominant amino acids in the selected NS1 mutant positions from different hosts are shown in Figure 1D.

### 3.2. NS1 Mutations Could Alter Viral Replicative Efficiency In Vitro

To evaluate the contribution of the above identified NS1 mutations on virus replication in vitro, we reintroduced the NS1 mutations into the WSN NS gene and rescued the NS1 mutant viruses in the WSN backbone. Viral growth kinetics were tested in MDCK and A549 cells. The data showed that the rNS1 F9Y, N4T + L95H + E172A and D2I + F9Y significantly enhanced virus titers at the early stages of infection 12 (*p* < 0.001) and 24 hpi (*p* < 0.01) compared to wild type WSN in MDCK cells (Figure 2A).

The rNS1 F9Y, N4T + L95H + E172A slightly improved viral replication at 36 and 48 hpi. Particularly, rNS1 F9Y increased viral titers 5.5- and 3.6-fold compared to wild type at 12 and 24 hpi, respectively. The rNS1 harboring A82V, L95P, K221E, T58S + I145M mutations have little or no effect on virus replication at all examined time points. Surprisingly, rNS1 D2I + T58S resulted in viral growth curves significantly lower than WSN at 12–60 hpi in MDCK cells (*p* < 0.05, 0.01, 0.01, ns, 0.001). In addition, rNS1 F9Y had a significantly increased ability to grow in A549 cells compared to the WSN virus at 12–72 hpi (*p* < 0.0001, *p* < 0.0001, *p* < 0.01, *p* < 0.05, *p* < 0.05, *p* < 0.05) (Figure 2B). The rNS1 K221E increased the viral replication titer at 12, 24 and 72 hpi (*p* < 0.01, *p* < 0.05, *p* < 0.05). However, rNS1 D2I + T58S dramatically attenuated the viral replicative efficiency in A549 cells (*p* < 0.0001, *p* < 0.001, *p* < 0.0001, *p* < 0.0001, *p* < 0.001, *p* < 0.0001). Taken together, among the selected NS1 mutations, rNS1 F9Y facilitated viral replication in vitro, while other NS1 mutant viruses displayed a close or lower replicative ability in MDCK and A549 cells compared to WSN.

### 3.3. In Vivo Infection Characterizations of NS1 Mutant Viruses

To further understand the effect of the screened NS1 mutations on replication and virulence in mice, BALB/c mice were intranasally infected with 5 × 10^3^ PFU of WSN wild type and NS1 mutant viruses. Mice body weight loss results indicated that the rNS1 F9Y mutant induced significantly more weight loss than the WSN wild type and the infected mice reached a maximum average body weight loss of 15.2% at 8 dpi. In addition, mice infected with rNS1 L95P and K221E showed a body weight decrease trend compared to the WSN infected group. Nevertheless, the following mutant viruses rNS1 A82V, rNS1 D2I + F9Y, rNS1 D2I + T58S, rNS1 T58S + I145M and rNS1 N4T + L95H + E172A did not result in a significant decrease in body weight (Figure 3A). Mice infected with WSN wild type virus did not result in any mortality throughout the course of the experiment (Figure 3B). In contrast, the NS1 gene of WSN possessing the F9Y, L95P or K221E mutations affected the mortality of the infected mice ranging from 20% to 40%. The rNS1 F9Y remained the most virulent virus in all eight recombinant viruses, whereas the other five mutants did not result in any mortality throughout the course of the experiment (Figure 3B). In addition, we measured the ratio of lung to body weight for the infected mice at 3 and 5 dpi with 10^4^ PFU of WSN wild type and NS1 mutant viruses. The rNS1 F9Y showed a higher ratio of lung to body weight compared to the WSN infected group on 3 dpi (*p* < 0.05) (Figure 3C), which suggested a severe pathological change in the lungs. The remaining NS1 mutants did not show any significant ratio of lung to body weight difference compared to the WSN wild type on 3 and 5 dpi (Figure 3D). To compare the virus replication properties in mice, we next infected BALB/c mice intranasally with the various NS1 mutants and the viral titers of infected mice lungs were determined on 3 and 5 dpi (Figure 3E,F). The results showed that the mutant rNS1 F9Y increased replication in mice lungs two-fold higher than that of the WSN wild type on 3 dpi. The remaining NS1 mutants resulted in a decreased replicative ability in mice lungs at both 3 and 5 dpi. Particularly, the virus comprising the NS1 D2I + T58S mutation induced a significantly low viral titer in mice lungs in comparison with the WSN virus on 3 and 5 dpi (*p* < 0.01, *p* < 0.001). Moreover, a pathological analysis showed that the WSN infected mouse lung displayed moderate inflammation along with immune infiltration. The lungs of rNS1 F9Y, L95P and K221E mutant infected mice showed severe inflammatory cell infiltration and more pathological lesions (Figure 3G). Together, the in vivo data demonstrated that the F9Y mutation in NS1 contributed to increased pathogenicity, while D2I + T58S mutations reduced virulence in mice.

### 3.4. The NS1 Mutant Viruses Contributed to IFN Expression in Mice Lungs

NS1 is the major influenza viral IFN antagonist. To assess the effect of the NS1 mutations on the ability of the virus to downregulate IFN-β production, the mRNA relative expressive level of IFN-β in virus infected mice lungs was detected by qPCR (Figure 4A,B). The results showed that rNS1 F9Y was the sole mutant to induce significant downregulation of IFN-β production compared to the WSN wild type virus at 3 dpi (*p* < 0.05). However, the IFN-β levels in the rNS1 F9Y infected lungs at 5 dpi showed no difference compared to the wild type WSN. The mutants rNS1 A82V, L95P, K221E, D2I + F9Y, D2I + T58S and N4T + L95H + E172A induced comparable levels of IFN-β in the mice lungs at 3 and 5 dpi compared to WSN. However, the rNS1 T58S + I145M virus showed a higher expression level of IFN-β level compared to the WSN at both 3 and 5 dpi (*p* < 0.01, *p* < 0.01), which correlated with its low replicative efficiency in mice lungs (Figure 3E,F). We thus concluded that rNS1 F9Y was capable of antagonizing IFN-β production in the mouse lung and, therefore, enhancing the viral replication in the early stages of virus infection. However, the NS1 mutant T58S + I145M demonstrated a reduced ability to suppress IFN-β production both at 3 and 5 dpi. All the remaining NS1 mutants are comparable to WSN WT in their ability to antagonize IFN-β production in the lungs in vivo.

### 3.5. NS1 F9Y Mutation Inhibited Expression of Genes Involved in Cytokine Responses In Vitro

The above data suggested that the NS1 F9Y mutation could significantly increase virus replication and pathogenicity in vitro and in vivo. To further understand the underlined mechanisms by which rNS1 F9Y enhances viral fitness, we first used a luciferase reporter gene assay to determine the ability of NS1 F9Y to suppress IFN-β transcription. As shown in Figure 5A, compared with the WSN NS1 protein, the NS1 F9Y significantly inhibited IFN-β synthesis (*p* < 0.05). To further clarify whether the rNS1 F9Y virus affected host innate immune responses in vitro, A549 cells were infected with WSN and rNS1 F9Y viruses. At 6 and 12 hpi, the mRNA levels of key innate immune molecules IL6, IL28, IL29, IFNα, IFNβ, ISG15, ISG20, MX1, TNFα, IFIT1, IFIT2, IFIT3, IFITM, MDA5, TLR7, RIGI, TLR3, TRIM22, TRIM25 and OASL were measured by qPCR (Figure 5B). The mRNA expression levels of IL29, IFNα, IFNβ, ISG15, IFIT1, IFIT2, IFIT3, TLR7, RIGI, TRIM25 and TLR3 mRNA induced by the rNS1 F9Y influenza virus were remarkably suppressed compared to those induced by WSN at 6 hpi. Similarly, in comparison to the WSN infection group, the expression levels of IL29, IFNα, IFNβ, MX1, IFIT1, IFIT2, IFIT3, TLR7, RIGI, TRIM22, TRIM25 and TLR3 in rNS1 F9Y infected cells were significantly inhibited at 12 hpi. These results suggested that the F9Y mutation in NS1 inhibited the transcription of genes related to host innate immunity.

### 3.6. NS1 D2I Mutation Inhibited Viral Growth and Upregulated Cytokine Expression In Vitro

We have confirmed that the rNS1 D2I + T58S virus inhibited viral replication ability both in vivo and in vitro. To further demonstrate the roles of the individual mutations D2I and T58S on viral growth, we generated mutant viruses containing single D2I and T58S mutations in a WSN background. Firstly, the growth kinetics analysis of the single and double sites mutated viruses were tested in MDCK and A549 cells. The results showed that the individual NS1 T58S mutation could slightly decrease virus replication compared to the WSN virus in both MDCK and A549 cells at all tested time points (Figure 6A,B).

However, the virus titers of the double mutant NS1 D2I and T58S further dropped in both MDCK and A549 cells compared to those of the WSN and single mutant rNS1 T58S viruses. Interestingly, the single NS1 D2I mutation significantly decreased the replicative ability of the WSN virus in these two mammalian cell lines. In comparison with the WSN virus, the rNS1 D2I virus reduced viral yield by 28.2- and 25.8-fold in MDCK and A549 cells at 36 hpi. Moreover, the expression profiles of the key cytokines in WSN and rNS1 D2I infected A549 cells were assessed at the transcriptional level by qPCR. The results showed that the mRNA levels of cytokines (including IL6, IL28, IL29, IFNα, IFNβ, ISG15, MX1 and TNFα) were significantly upregulated in A549 cells infected with the rNS1 D2I virus at 6 hpi compared to the WSN infected cells (*p* < 0.01, *p* < 0.001, *p* < 0.001, *p* < 0.001, *p* < 0.001, *p* < 0.001, *p* < 0.0001, *p* < 0.001) (Figure 6C). Taken together, we concluded that the NS1 D2I mutation contributed to the attenuated viral replication in infected cells as well as the upregulated expression of cytokines.

### 3.7. NS1 D2I Mutation Decreased the Pathogenicity of the WSN Virus in Mice

We next tested the pathogenicity of the NS1 D2I mutant virus in a mice model.

The body weight loss results indicated that 10^4^ PFU of the WSN virus caused the most severe disease expressed by body weight loss compared with the rNS1 D2I and rNS1 D2I + T58S viruses. The rNS1 D2I + T58S mutant resulted in a moderate body weight change in infected mice. However, the mice infected with the rNS1 D2I virus only lost approximately 4.6% of their initial body weight at 7 dpi and then recovered right away (Figure 7A). As shown in Figure 7B, all of the mice infected with WSN died at 8 dpi. Whereas all of the mice infected with the rNS1 D2I virus survived throughout the 14 monitored days. In addition, to assess the viral replication in mice lungs, BALB/c mice were intranasally infected with the rNS1 mutants or WSN viruses and the mice lungs were then collected for virus shedding analysis. We found that the viral titers in the WSN infected group were >10-fold and >400-fold higher than the rNS1 D2I infected group at 3 and 5 dpi, respectively. The double mutant rNS1 D2I + T58S virus replicated more efficiently than the rNS1 D2I, but to a lower titer than the WSN virus in 3 and 5 dpi (Figure 7C,D). The cytokines and chemokines expression profiles indicated that although the WSN virus replicated efficiently in mice lungs, the transcript levels of IL-4 and IL6 in the NS1 F9Y mutant virus infection group were about 6.7- and 3.5-fold higher than those in the WSN infection group at 3 dpi (*p* < 0.01, *p* < 0.05) (the control group were PBS-infected mice). The expression of MIP-1α, MCP-1 and TNFα showed no difference between the WSN and rNS1 D2I viruses (Figure 7E). The above results suggested that the D2I mutation in NS1 significantly reduced the virulence of the WSN virus in mice.

### 3.8. The Effect of NS1 Mutations F9Y and D2I on Nuclear Export Effectivity

The gene segment 8 of IAV includes both unspliced and spliced mRNA transcripts, encoding NS1 and NEP proteins. In addition, the two proteins share the first 10 N-terminal amino acids. To verify whether D2I and F9Y mutations in the NS1 could also affect the nuclear export function of NEP, we constructed the protein expression plasmids that expressed eGFP-NEP, eGFP-NEP-D2I and eGFP-NEP-F9Y, then transfected these plasmids into HEK293T cells. Subcellular localization of eGFP fusion proteins were examined via confocal microscopy at 24 h post-transfection (Figure 8A). We found that there was no significant difference in nuclear export efficiency between the wild type NEP and mutant NEP proteins. The ratio of IntDEN within nuclear to cytoplasmic demonstrated nuclear export effectivity were at a comparable level (Figure 8B). Therefore, the results indicated that the D2I and F9Y mutations in NEP had no influence on the effectivity of NEP nuclear export and did not further affect viral replication efficiency.

## 4. Discussion

The replication and pathogenesis of IAV depends on the interaction between viral proteins and the host immune system. To replicate successfully in the host, IAVs have evolved to evade host immune response by encoding proteins such as NS1, which mitigate host antiviral responses and facilitate viral replication. To identify novel mutations in the NS1 protein that antagonize the host immune response, in this study, we constructed a random mutant plasmid library and then rescued a NS1 gene random mutant virus library in a WSN background. Through lung-to-lung passage in mice, a cluster of NS1 mutant viruses were selected. A comparison with the NS1 sequences of human, swine and avian H1N1 natural isolates proved that the NS1 mutations selected in our study were previously selected in nature and are, therefore, convergent mutations.

To avoid the effect of additional adaptive mutations in other IAVs genes acquired during viral passaging in mice lungs, we then cloned only the identified NS1 mutations into WSN NS1 gene to generate the NS1 mutant viruses. A series of in vitro and in vivo studies were performed to assess the functions of the NS1 mutations. We found that the rNS1mut F9Y virus replicated in MDCK and A549 cells to a higher titer than that of the wild type WSN virus. Analysis of the role of F9Y in NS1 in the mice model demonstrated an increased virulence and disease severity. Inhibition of interferon-induced cellular signals plays a critical role in influenza infection. To clarify how the rNS1 F9Y virus affected host innate immune responses in vitro, a double luciferase assay was performed to examine the regulatory effect of the NS1 protein on the IFN promoter, the results suggested that rNS1mut F9Y suppressed IFN-β synthesis more efficiently than the WSN virus. Moreover, the amount of IFN-β produced in rNS1 F9Y infected mouse lung was significantly less than the WSN infected mice. The pathology changes of mice lungs caused by rNS1 F9Y virus was more severe than that caused by WSN. We know that NS1 mainly inhibits IFN induction via sequestration of dsRNA by interacting with the RNA binding domain of NS1. TRIM family members are critical regulators of innate immune responses against microbial pathogens, including viruses [30]. During influenza virus infection, NS1 inhibits oligomerization of TRIM25 by interacting with the E3 ubiquitin ligase domain to prevent ubiquitination of RIG-I K63, thereby blocking formation of the RIG-I-MAVS signal complex and effectively inhibiting production of IFN [31]. In our study, rNS1 F9Y virus clearly enhanced the inhibition of IFN induction, as determined by qPCR analysis of mRNA expression levels. A549 cells infected with rNS1 F9Y virus significantly downregulated the expression of mRNA encoding TRIM25 and RIG-I compared to that infected with WSN. We speculated that the rNS1 F9Y mutant may inhibit the expression of TRIM25, block the RIG-I pathway, and inhibit the expression of IFN. In addition, the level of IL-29 in A549 cells was inhibited significantly following infection with the rNS1mut F9Y virus, indicating that the F9Y mutation in WSN NS1 could indeed inhibit host immune responses during the early stage of viral infection. The observation that the NS1 F9Y mutation was under positive selection and therefore adaptive was evident by its increased prevalence in mouse lung passages (Appendix A). This indicated that the F9Y mutation conferred a selective advantage relative to the wild type NS1 genes possessing Phenylalanine at position 9.

In addition, NS1 with a D2I mutation significantly decreased the virus replication in cells and attenuated its pathogenicity in mice. We further found that the expression levels of some cytokines and chemokines stimulated by rNS1mut D2I virus infection were upregulated, which may explain its reduced replicative efficiency in the cells and mice model. The NEP protein has also been linked to the regulation of viral transcription and replication of the viral genome in a mechanism that is independent of vRNP nuclear export [32]. Since both D2I and F9Y mutations are localized in the overlapping regions of NS1 and NEP proteins, the differences of virus replicative ability and pathogenicity in wild type and NS1 mutants may be caused by the functional change of NEP protein. We next verified that the nuclear export efficiency of NEP wt, NEP D2I and NEP F9Y did not show any significantly difference, which indicated that the biological functions of NS1 D2I and F9Y were independent of their NEP mutations.

In summary, we identified two novel NS1 mutations isolated from a random mutant virus library that could remarkably affect the replicative ability and virulence of A/WSN/1933 virus in vitro and in vivo. Our data provide additional evidence that NS1 protein is a critical virulence factor and that the two NS1 N-terminal mutations may be served as potential targets for antiviral drug design and attenuated vaccine development.

## Figures and Tables

**Figure 1 viruses-14-01206-f001:**
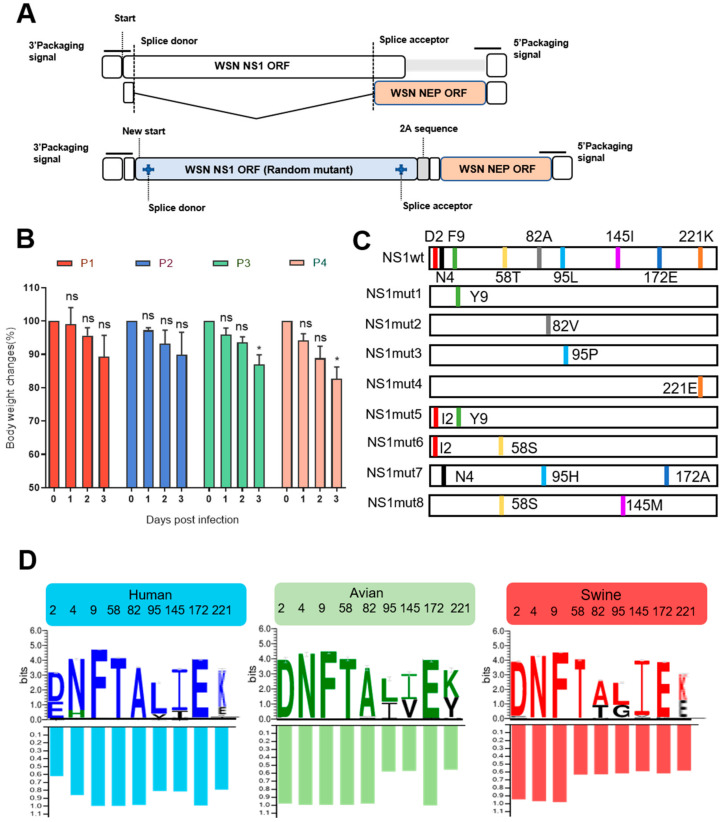
Generation and characterization of the NS1 mutants from the lung-to-lung passaged virus library. (**A**) Schematic representation of the wild type NS segment and the NS1-2A-NEP linear segment. The splice donor (SD) and acceptor site (SA) of the linear NS segment were mutated to prevent gene splicing. NS1 and NEP are co-translationally separated by the recoding activity of the 2A peptide of porcine teschovirus 1 (PTV-1). (**B**) Lung-to-lung passage of library virus in mice. Six to eight-week-old BALB/c mice were infected intranasally with the NS1 random mutant library viruses. The lungs were collected and homogenized on 3 dpi. The resulting supernatants were used for the next round infection. The illustrations show the mice body weight changes in each passage. *p* values were calculated by using the Student’s *t*-test, ns = not significant, * *p* < 0.05 (**C**) The identified NS1 mutations from purified lung-to-lung passage samples. (**D**) Prevalence and proportion of amino acids at selected NS1 mutation residues in different hosts of H1N1 IAVs. The logo plot represents the prevalent amino acid at each NS1 mutant site from human, avian and swine H1N1 influenza virus. The histogram shows the predominant amino acids frequency at every selected NS1.

**Figure 2 viruses-14-01206-f002:**
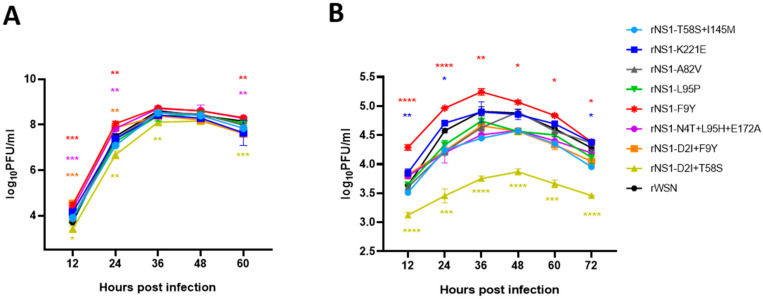
Growth kinetics of NS1 mutants in cells. (**A**) Growth kinetics of NS1 mutants in MDCK cells. (**B**) Growth kinetics of NS1 mutants in A549 cells. *p* values were calculated by using the Student’s *t*-test, * *p* < 0.05, ** *p* < 0.01, *** *p* < 0.001, **** *p* < 0.001.

**Figure 3 viruses-14-01206-f003:**
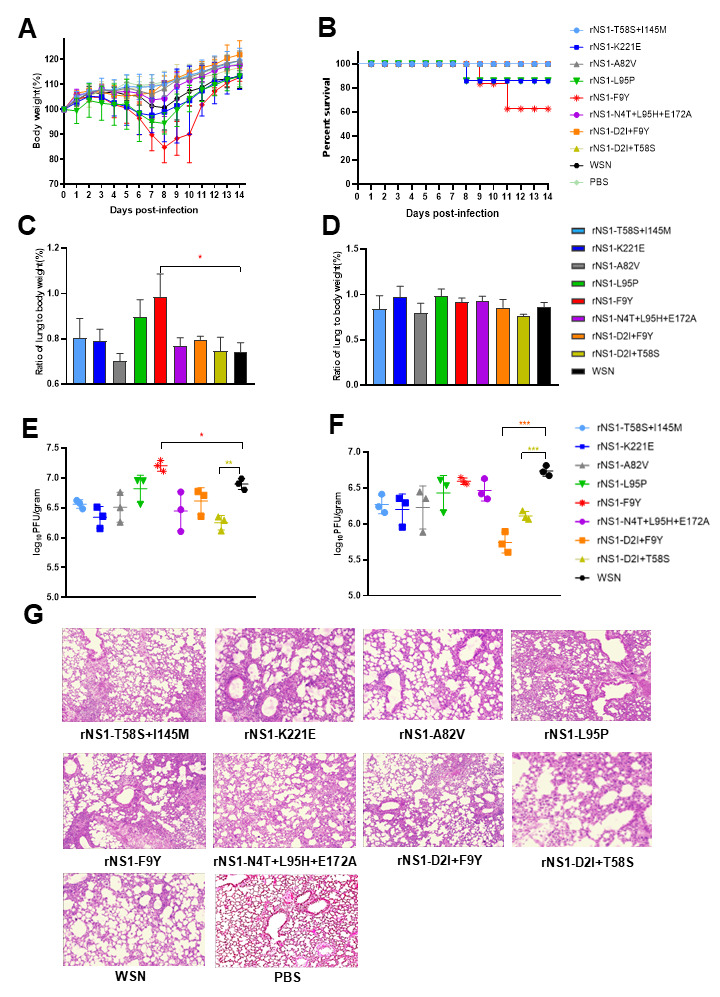
Replication and pathogenicity of NS1 mutants in mice. (**A**) Body weight changes of mice infected with 5 × 10^3^ PFU of rNS1 mutants and WSN viruses. (**B**) Survival curves of mice infected with 5 × 10^3^ PFU of rNS1 mutants and WSN viruses. Mice were euthanized and scored dead when the mice lost 25% of their baseline body weight (measured on the day of infection). (**C**) Ratio of lung to body weight of rNS1 mutants and WSN viruses at 3 dpi. BALB/c mice inoculated with 10^4^ PFU of rNS1 mutants and WSN viruses, the ratios of lung to body weight were calculated. (**D**) Ratio of lung to body weight of rNS1 mutants and WSN viruses at 5 dpi. (**E**) Viral replication titers of the infected mice lungs at 3 and (**F**) 5 dpi. (**G**) Pathological analysis of mice lungs infected with rNS1 mutants and WSN viruses at 3 dpi. *p* values were calculated by using the Student’s *t*-test, * *p* < 0.05, ** *p* < 0.01, *** *p* < 0.001.

**Figure 4 viruses-14-01206-f004:**
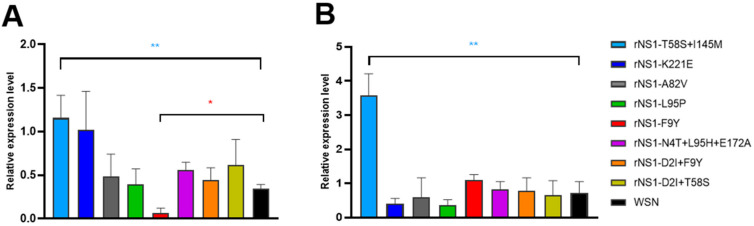
The relative mRNA expression levels of IFN induced by rNS1 mutants and WSN viruses in mice. (**A**) Lungs were collected at 3 dpi (**B**) and 5 dpi for IFNβ mRNA detection by qPCR. Expression levels were normalized to that of GAPDH and are presented as changes (n-fold) in induction relative to the control value. *p* values were calculated by using the Student’s *t*-test, * *p* < 0.05, ** *p* < 0.01.

**Figure 5 viruses-14-01206-f005:**
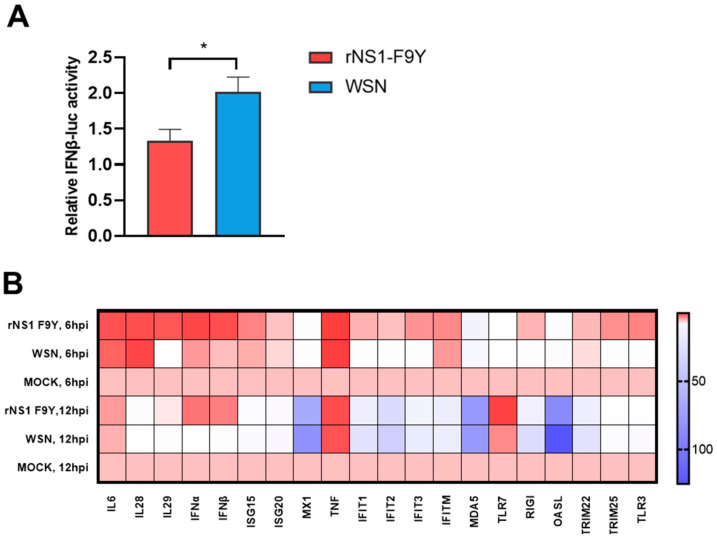
The NS1 F9Y mutation impaired host innate immune responses. (**A**) The ability of the NS1 F9Y protein to inhibit IFN-β promoter activation. Human 293T cells were transiently co-transfected with a reporter plasmid encoding firefly luciferase gene under the control of the IFN promoter, a plasmid constitutively expressing Renilla luciferase gene under the control of the HSV-TK promoter (PRL-TK), and the pCAGGS-WSN NS1 or pCAGGS-WSN NS1 F9Y mutant plasmids (* *p* < 0.05). (**B**) Heat map showing the differential expression of cytokine encoding genes in virus infected A549 cells. A549 cells were infected with WSN and rNS1 mut F9Y at a MOI of 1. The cells were collected at 6 and 12 hpi, and the cytokine genes expression levels were quantified by qPCR.

**Figure 6 viruses-14-01206-f006:**
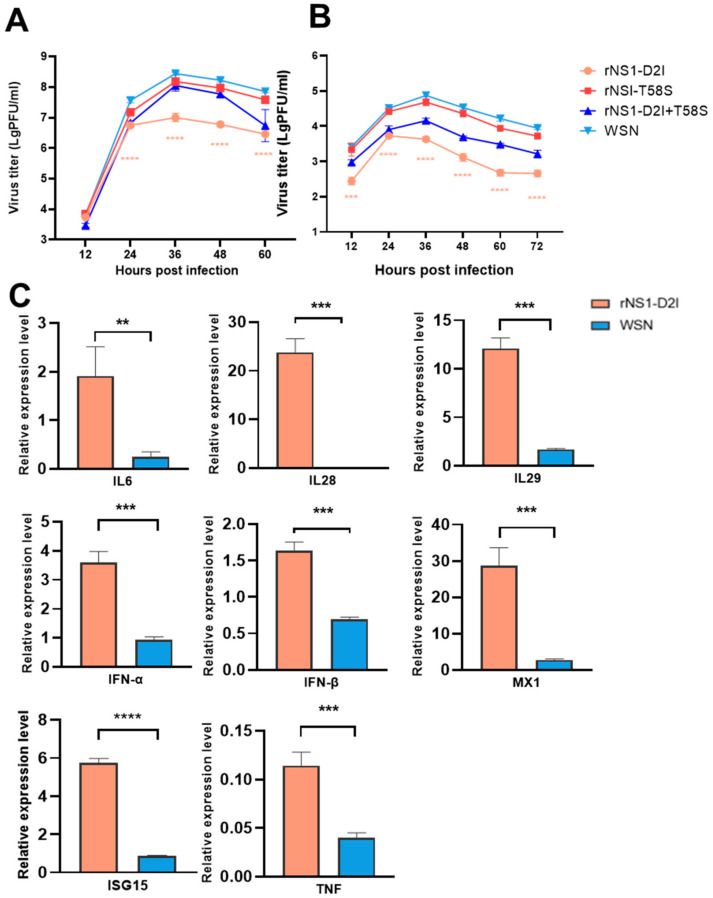
In vitro characterizations of WSN NS1 D2I mutated viruses. (**A**) Growth kinetics of rNS1 D2I mutant in MDCK cells. (**B**) Growth kinetics of rNS1 D2I mutant in A549 cells. (**C**) The mRNA expression levels of IFNs and ISGs in infected A549 cells. *p* values were calculated by using the Student’s *t*-test, ** *p* < 0.01, *** *p* < 0.001, **** *p* < 0.001.

**Figure 7 viruses-14-01206-f007:**
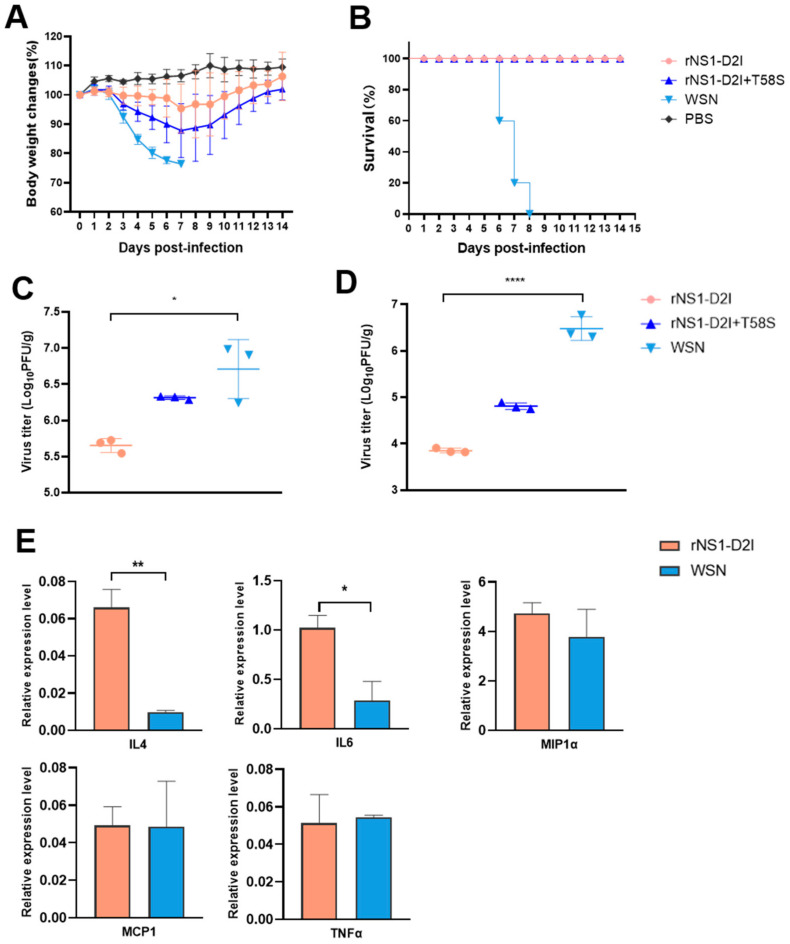
In vivo characterizations of WSN NS1 D2I mutated virus. (**A**) Body weight changes of mice infected with 10^4^ PFU of rNS1 D2I, rNS1 D2I + T58S and WSN viruses. (**B**) Survival curves. (**C**) Viral titers in infected mice lungs on 3, and (**D**) 5 dpi. (**E**) The mRNA expression levels of IL-4, IL6, MIP-1α, MCP-1 and TNFα in BALB/c mice infected with WSN and rNS1 D2I mutant. Infected mice lungs were collected at 3 dpi for cytokine detection by qPCR. Expression levels were normalized to that of GAPDH and are presented as changes (n-fold) in induction relative to the control value. *p* values were calculated by using the Student’s *t*-test, * *p* < 0.05, ** *p* < 0.01, **** *p* < 0.001.

**Figure 8 viruses-14-01206-f008:**
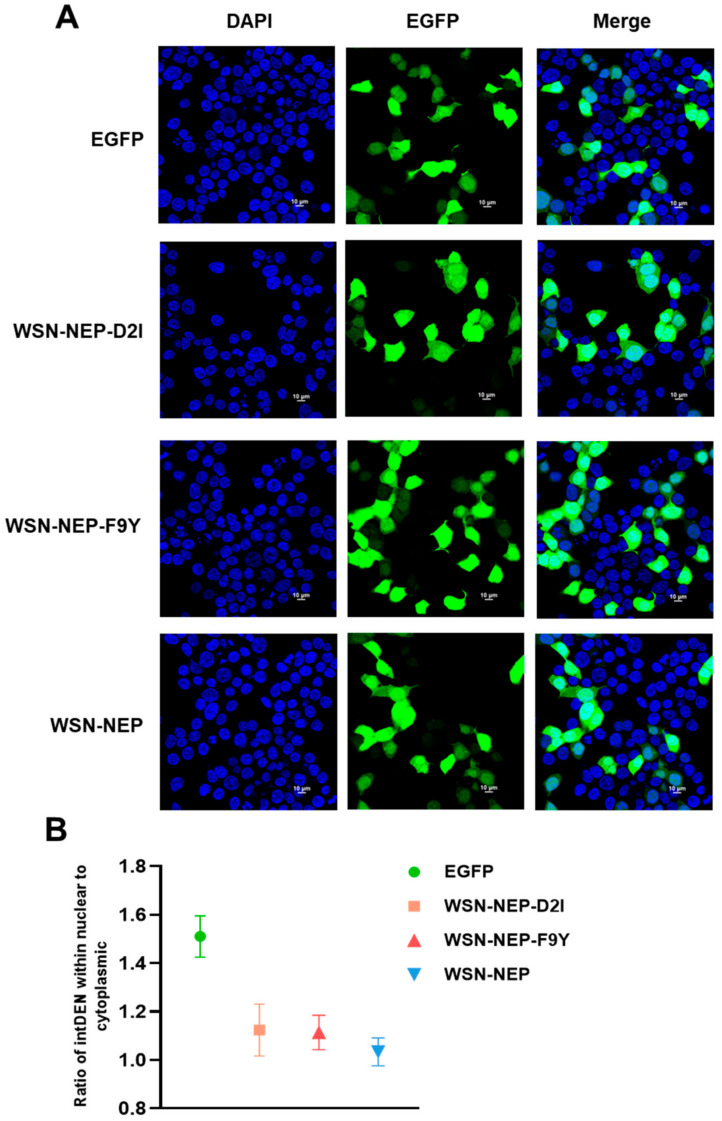
Identification of the WSN-NEP and NEP mutants mediating the nuclear export of eGFP fusion protein. (**A**) HEK293T cells growing on cover glass were transfected with pCAGGS encoding eGFP-NEP, eGFP-NEP D2I mutant or eGFP-NEP F9Y mutant. Cells were stained with the 4, 6-diamidino-2-phenylindole (DAPI) and were examined via confocal microscopy 24 h post-transfection. (**B**) Quantification of fluorescence signal was performed by ImageJ.

## Data Availability

Not applicable.

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
