# Peer review of "D2I and F9Y Mutations in the NS1 Protein of Influenza A Virus Affect Viral Replication via Regulating Host Innate Immune Responses"

_viruses, 2022, doi:10.3390/v14061206_

Round 1
Reviewer 1 Report
The submitted manuscript presents interesting data and relies on an experimental design that merits attention. Most of the conclusions are supported by valid experimental evidence. However, the results from Fig. 2 do no support the conclusions and the conclusions regarding the differential growth properties of the mutant viruses should be toned down. Indeed, the only mutant showing a differential growth compared to the WT is F9Y. The other mutants don’t and the text should be modified accordingly. Please modify the text.
The other major comment regarding the manuscript concerns the Material and Methods. Indeed, much more detail is needed to allow the readers to understand what has been down and to assess if the conclusions are valid or not. The authors should provide a detailed protocol on the random mutagenesis PCR, on the cloning of mutagenized sequence and on the number of mutations obtained per segment (including statistical analyses used to evaluate the average number of mutations).
In addition, it is unclear if the viruses generated to measure differential growth in cell culture and in vivo were generated using the NS-linear plasmid or the “normal” NS segment. Please clarify.
In addition, the authors need to clarify the following points:
- Describe Fig 1D in Material and Methods and also in the legends: the figure is not understandable in its current state
- Not clear in the text and legend which dose of virus was used in Fig 7. Why WT is so pathogenic in Fig 7 compared to Fig 3?
- IntDEM Fig 8 : what does this mean? Describe the methods
Finally, English editing is required. I can’t list everything but here are a few examples:
- Line 22: “While” is not adequate
- Line 451: “As expected” is not explained
- Line 478: “we confirmed” does not make sense, or you should state what you confirm and provide a reference
Author Response
Dear Editor,
I with my co-authors, appreciate for your letter on our manuscript entitled “D2I and F9Y Mutations in the NS1 Protein of Influenza A Virus Affect Viral Replication Via Regulating Host Innate Immune Responses”. Thank you very much for giving us an opportunity to revise our manuscript. We are also very grateful to the reviewers for their valuable comments by cautious revision, which helps to improve our manuscript. Based on reviewer’s suggestions, we answered each question one by one. We have made careful revision on the original manuscript, all revised portions marked in yellow in the revised manuscript.
Most sincerely, Corresponding author
Jihui Ping
May. 25 2022
Please see our responses as follows:
Reviewer(s)' Comments to Author:
Reviewer1:
Comments to the Author
The submitted manuscript presents interesting data and relies on an experimental design that merits attention. Most of the conclusions are supported by valid experimental evidence. However:
1.The results from Fig.2 do no support the conclusions and the conclusions regarding the differential growth properties of the mutant viruses should be toned down. Indeed, the only mutant showing a differential growth compared to the WT is F9Y. The other mutants don’t and the text should be modified accordingly. Please modify the text.
Response: Thank you for your valuable advice. We have modified this section in the revised manuscript as suggested (Page 7, lines 288-290).
2.The other major comment regarding the manuscript concerns the Material and Methods. Indeed, much more detail is needed to allow the readers to understand what has been down and to assess if the conclusions are valid or not. The authors should provide a detailed protocol on the random mutagenesis PCR, on the cloning of mutagenized sequence and on the number of mutations obtained per segment (including statistical analyses used to evaluate the average number of mutations).
Response: Thank you for your valuable suggestion. We have supplemented a detailed protocol on the random mutagenesis PCR in Materials and Methods section in the revised manuscript as suggested (Page 3, lines 111-117).
3.In addition, it is unclear if the viruses generated to measure differential growth in cell culture and in vivo were generated using the NS-linear plasmid or the “normal” NS segment. Please clarify.
Response: Thank you for your valuable question. The NS1 mutant viruses were generated based on “normal” NS segment. This section was added in the revised manuscript (Page 2, lines 98).
4.In addition, the authors need to clarify the following points:
Describe Fig 1D in Material and Methods and also in the legends: the figure is not understandable in its current state.
Response: Thank you very much for your question. This section was rewritten in the revised manuscript (Page 6, lines 255-259).
Not clear in the text and legend which dose of virus was used in Fig 7. Why WT is so pathogenic in Fig 7 compared to Fig 3?
Response: Thank you for your valuable question. The infection dose of WSN or NS1 mutant viruses used in Fig7 was 104 PFU which was two-fold higher that of in figure 3A and 3B. This section was rewritten in the revised manuscript (Page 13, lines 411-412; Page13, lines 418).
- IntDEM Fig 8 : what does this mean? Describe the methods
Response: Thank you for your valuable question. The word “IntDEM” was rewritten as “IntDEN” in the revised manuscript. The ratio of IntDEN (Integrated density) within nuclear to cytoplasmic indicates nuclear export activity of NS2 protein. The additional description of method has been added to the revised manuscript (Page 5, lines 211-213).
5.Finally, English editing is required. I can’t list everything but here are a few examples:
- Line 22: “While” is not adequate
Response: Thank you for your language advice, the words “While” in page 1 lines 24 had been deleted. And uploaded with the new version of manuscript.
- Line 451: “As expected” is not explained
Response: Thank you for your suggestion. The words “As expected” had been replaced with “We found” (Page 16, lines 471).
- Line 478: “we confirmed” does not make sense, or you should state what you confirm and provide a reference
Response: Thank you for your advice, the words “we confirmed that” in page 16 lines 499 had been deleted. And uploaded with the new version of manuscript.

Reviewer 2 Report
The authors found that D2I and F9Y mutations in the NS1 protein of H1N1 influenza virus (A/WSN/1933)affected viral replication. Their results showed that NS1 F9Y mutation in the WSN virus significantly enhanced viral replication in cells and increased virulence in BALB/c mice, but NS1 D2I mutation attenuated the viral replication and pathogenicity in both in vitro and in vivo models. The NS1 F9Y mutant virus exhibited systematic and selective inhibition of cytokine responses as well as inhibited expression of IFN. Therefore, the two NS1 mutations may serve as potential targets for antiviral drugs as well as attenuated vaccine development. This study is interesting and some concerns are described below.
1) Abstract: change “Further studies demonstrated that the NS1 F9Y mutant exhibited systematic and selective inhibition of cytokine responses as well as inhibit expression of IFN” to “Further studies demonstrated that the NS1 F9Y mutant virus exhibited systematic and selective inhibition of cytokine responses as well as inhibited expression of IFN”.
2) Line 93, change “its NS1 mutants were generated by reverse genetics…” to “its NS1 mutant viruses were generated by reverse genetics…”.
3) Line 221-222, change “BALB/c mice were lung-to-lung passaged with the library virus for 4 rounds.” to “the library viruses were lung-to-lung passaged in BALB/c mice for 4 rounds.”.
4)Line 227-228, change “Sequencing results in respect to the NS1 gene showed that in total 13 227 mutant viruses were identified…” to “Sequencing results in respect to the NS1 gene showed that 13 mutant viruses were identified…”.
5)Line 227-228, change “Sequencing results in respect to the NS1 gene showed that in total 13 227 mutant viruses were identified…” to “Sequencing results in respect to the NS1 gene showed that 13 mutant viruses were identified…”.
6)Line 306, change “reduce” to “reduced”.
Author Response
Dear Editor,
I with my co-authors, appreciate for your letter on our manuscript entitled “D2I and F9Y Mutations in the NS1 Protein of Influenza A Virus Affect Viral Replication Via Regulating Host Innate Immune Responses”. Thank you very much for giving us an opportunity to revise our manuscript. We are also very grateful to the reviewers for their valuable comments by cautious revision, which helps to improve our manuscript. Based on reviewer’s suggestions, we answered each question one by one. We have made careful revision on the original manuscript, all revised portions marked in yellow in the revised manuscript.
Most sincerely, Corresponding author
Jihui Ping
May. 25 2022
Please see our responses as follows:
Reviewer(s)' Comments to Author:
Reviewer2:
Comments to the Author
The authors found that D2I and F9Y mutations in the NS1 protein of H1N1 influenza virus (A/WSN/1933)affected viral replication. Their results showed that NS1 F9Y mutation in the WSN virus significantly enhanced viral replication in cells and increased virulence in BALB/c mice, but NS1 D2I mutation attenuated the viral replication and pathogenicity in both in vitro and in vivo models. The NS1 F9Y mutant virus exhibited systematic and selective inhibition of cytokine responses as well as inhibited expression of IFN. Therefore, the two NS1 mutations may serve as potential targets for antiviral drugs as well as attenuated vaccine development. This study is interesting and some concerns are described below.
1.Abstract: change “Further studies demonstrated that the NS1 F9Y mutant exhibited systematic and selective inhibition of cytokine responses as well as inhibit expression of IFN” to “Further studies demonstrated that the NS1 F9Y mutant virus exhibited systematic and selective inhibition of cytokine responses as well as inhibited expression of IFN”.
Response: Thank you for your language suggestion, we have fixed this mistake in the revised manuscript as “Further studies demonstrated that the NS1 F9Y mutant virus exhibited systematic and selective inhibition of cytokine responses as well as inhibited expression of IFN”. (In page 1, lines 25-27).
2.Line 93, change “its NS1 mutants were generated by reverse genetics…” to “its NS1 mutant viruses were generated by reverse genetics…”.
Response: Thank you for your language advice. The sentence “its NS1 mutants were generated by reverse genetics…” were rewritten as “its NS1 mutant viruses were generated by reverse genetics…” in the revised manuscript (Page 2, line 98).
3.Line 221-222, change “BALB/c mice were lung-to-lung passaged with the library virus for 4 rounds.” to “the library viruses were lung-to-lung passaged in BALB/c mice for 4 rounds.”.
Response: Thank you for your language advice. In Page 5, line 235-236 of revised manuscript, the sentence “BALB/c mice were lung-to-lung passaged with the library virus for 4 rounds” was rewritten as “the library viruses were lung-to-lung passaged in BALB/c mice for 4 rounds”.
4.Line 227-228, change “Sequencing results in respect to the NS1 gene showed that in total 13 mutant viruses were identified…” to “Sequencing results in respect to the NS1 gene showed that 13 mutant viruses were identified…”.
Response: Thank you for your language advice. In Page 5 line 241-242 of revised manuscript, the sentence “Sequencing results in respect to the NS1 gene showed that in total 13 mutant viruses were identified…” had been replaced with “Sequencing results in respect to the NS1 gene showed that 13 mutant viruses were identified…”.
5.Line 227-228, change “Sequencing results in respect to the NS1 gene showed that in total 13 227 mutant viruses were identified…” to “Sequencing results in respect to the NS1 gene showed that 13 mutant viruses were identified…”.
Response: Thank you for your language advice. In Page 5 line 241-242 of revised manuscript, the sentence “Sequencing results in respect to the NS1 gene showed that in total 13 mutant viruses were identified…” had been replaced with “Sequencing results in respect to the NS1 gene showed that 13 mutant viruses were identified…”.
6.Line 306, change “reduce” to “reduced”.
Response: Thank you for your language advice. The word “reduce” had been replaced with “reduced” in the revised manuscript (Page 8, line 323).

Reviewer 3 Report
In the present study, Mengqi Yu and colleagues attempted to screen out the unidentified NS1 functional mutants. They found that NS1 F9Y mutation significantly enhanced viral growth in vitro and in vivo , while NS1 D2I mutation attenuated the viral replication and pathogenicity. Further studies demonstrated that the NS1 F9Y mutant inhibited the cytokine responses and expression of IFN. Overall, it is a meaningful and interesting work.
Major comments for consideration:
1. Some results about NS1 D2I mutation upregulated cytokine expression in vitro did not covered in the abstract.
2. The title of “Generation of NS1 random mutated plasmid library” does not match the contents.
3. The title of “The mutation F9Y in NS1 contributed to IFN induction in mice lung” is inconsistent with the content.
4. The Grammar needs to be modified carefully.
Minor comments for consideration:
1. Write the full name for the first abbreviation, such as pi.
2. Line 174, “Cells were lysed at 12 post-infection (hpi)” .
3. Figure 5.(A), the title of the ordinate need to be corrected.
4. Figure 8. (B) , WSN-NEP F9Y appeared two times, while WSN-NEP missed.
Author Response
Dear Editor,
I with my co-authors, appreciate for your letter on our manuscript entitled “D2I and F9Y Mutations in the NS1 Protein of Influenza A Virus Affect Viral Replication Via Regulating Host Innate Immune Responses”. Thank you very much for giving us an opportunity to revise our manuscript. We are also very grateful to the reviewers for their valuable comments by cautious revision, which helps to improve our manuscript. Based on reviewer’s suggestions, we answered each question one by one. We have made careful revision on the original manuscript, all revised portions marked in yellow in the revised manuscript.
Most sincerely, Corresponding author
Jihui Ping
May. 25 2022
Please see our responses as follows:
Reviewer(s)' Comments to Author:
Reviewer3:
Comments to the Author
In the present study, Mengqi Yu and colleagues attempted to screen out the unidentified NS1 functional mutants. They found that NS1 F9Y mutation significantly enhanced viral growth in vitro and in vivo , while NS1 D2I mutation attenuated the viral replication and pathogenicity. Further studies demonstrated that the NS1 F9Y mutant inhibited the cytokine responses and expression of IFN. Overall, it is a meaningful and interesting work.
Major comments for consideration:
1.Some results about NS1 D2I mutation upregulated cytokine expression in vitro did not covered in the abstract.
Response: Thank you for your valuable suggestion. This section was added in the revised manuscript (Page 1, lines 27-28).
2.The title of “Generation of NS1 random mutated plasmid library” does not match the contents.
Response: Thank you for your valuable suggestion. This section was modified in the revised manuscript (Page 3, lines 111-117).
3.The title of “The mutation F9Y in NS1 contributed to IFN induction in mice lung” is inconsistent with the content.
Response: Thank you for your valuable suggestion. The title of “The mutation F9Y in NS1 contributed to IFN induction in mice lung” was replaced with “The NS1 mutant viruses contributed to IFN expression in mice lung” This section was modified in the revised manuscript (Page 9, lines 334).
4.The Grammar needs to be modified carefully.
Minor comments for consideration:
1.Write the full name for the first abbreviation, such as pi.
Response: Thank you for your language suggestion, we have wrote the full name for the first abbreviation, and the manuscript has been thoroughly checked and revised the mistakes.
2.Line 174, “Cells were lysed at 12 post-infection (hpi)”.
Thank you for your language advice. In Page 4, line 180 of revised manuscript, the words “12 post-infection (hpi)” was rewritten as “12 hour post-infection (hpi)”.
3.Figure 5.(A), the title of the ordinate need to be corrected.
Thank you for your advice. The title of the ordinate in Figure 5.(A) have been corrected in revised manuscript.
4.Figure 8. (B) , WSN-NEP F9Y appeared two times, while WSN-NEP missed.
Response: Thank you for your valuable question. This section was modified in the figure 8B.

Reviewer 4 Report
The manuscript of Yu et al. describes a WSN backbone virus with NS1 F9Y mutation enhanced viral growth in MDCK and A549 cells, and pathogenicity in mice. Moreover, NS1 D2I mutation decreased the virus replication in cells and attenuated pathogenicity in mice. The changes of replication and pathogenicity of these NS1 mutant viruses were likely associated to ability of viral NS1 protein to suppress host innate immune responses such as induction of IFN-beta and other cytokines. I think the authors conducted sufficient experiments and provided persuasive data to support their conclusion. The result part seemed to be a little long but was described well to understand the study. It may need a further explanation at several parts of results (e.g., mouse experiment with D2I mutant and immunofluorescence).
To improve the manuscript, could the authors consider some specific points as following?
“Introduction”
Introduction section should be divided to several paragraphs adequately. It is same to Discussion section.
“Materials and Methods”
2.8 Mouse experiments
The inoculation doses of rWSN virus (5x10^3 PFU/50uL) and rNS1 viruses (10^4 PFU/50uL) was different. Did the 2-fold difference have no effect on the viral titer in lungs or weight loss of inoculated mice?
Was the inoculation dose for mouse experiment using D2I single mutant viruses was greater than that of F9Y experiment? Because the mice inoculated with WSN survived without weight loss for 14 days as shown in Figure 3A, but WSN-inoculated mice all died in Figure 7A.
Experimental infection of mice to examine cytokine mRNA expression was not described in Materials and Methods. The gene expression was shown in n-fold as relative expression level in Figure 7E. The control group is mock mouse (PBS-infected mice)?
“Results”
In Figure 5B, the expression level of TLR7 mRNA at 6 hpi in this heat map seemed to be same between WSN and NS1 F9Y. Both were higher than that of mock. Likewise, the expression levels of TRIM25 and TLR3 at 12 hpi looks same between WSN and NS1 F9Y. Alternatively, OASL expression level was different at 12 hpi. Those could not correspond to the description of results section (lines 349-354). Could you check them?
Could you please add the explanation of bars (rNS1-D2I and WSN) in Figure 6C (like Figure 7E).
At line 401, could you show the specific name of mutant viruses (D2I and D2I+T58S), instead of “the mutants”, to distinguish from the description of Figure 3.
At line 401, D21 is typo? D2I may be correct.
Could you explain about “IntDEM” described at line 427 and Figure 8B?
In Figure 8B, there were two WSN-NEP-F9Y and no WSN-NEP.
“Discussion”
At line 489, I don’t know which is better, NS2 mutations or NEP mutations.
Author Response
Dear Editor,
I with my co-authors, appreciate for your letter on our manuscript entitled “D2I and F9Y Mutations in the NS1 Protein of Influenza A Virus Affect Viral Replication Via Regulating Host Innate Immune Responses”. Thank you very much for giving us an opportunity to revise our manuscript. We are also very grateful to the reviewers for their valuable comments by cautious revision, which helps to improve our manuscript. Based on reviewer’s suggestions, we answered each question one by one. We have made careful revision on the original manuscript, all revised portions marked in yellow in the revised manuscript.
Most sincerely, Corresponding author
Jihui Ping
May. 25 2022
Please see our responses as follows:
Reviewer(s)' Comments to Author:
Reviewer4:
Comments to the Author
The manuscript of Yu et al. describes a WSN backbone virus with NS1 F9Y mutation enhanced viral growth in MDCK and A549 cells, and pathogenicity in mice. Moreover, NS1 D2I mutation decreased the virus replication in cells and attenuated pathogenicity in mice. The changes of replication and pathogenicity of these NS1 mutant viruses were likely associated to ability of viral NS1 protein to suppress host innate immune responses such as induction of IFN-beta and other cytokines. I think the authors conducted sufficient experiments and provided persuasive data to support their conclusion. The result part seemed to be a little long but was described well to understand the study. It may need a further explanation at several parts of results (e.g., mouse experiment with D2I mutant and immunofluorescence).
To improve the manuscript, could the authors consider some specific points as following?
“Introduction”
1.Introduction section should be divided to several paragraphs adequately. It is same to Discussion section.
Thank you for your valuable advice. The introduction and discussion section have been divided to several paragraphs adequately.
“Materials and Methods”
2.8 Mouse experiments
2.The inoculation doses of rWSN virus (5x10^3 PFU/50uL) and rNS1 viruses (10^4 PFU/50uL) was different. Did the 2-fold difference have no effect on the viral titer in lungs or weight loss of inoculated mice?
Thank you for your question. The mice infected dose of WSN or NS1 mutant viruses used in Figure 3A and 3B was 5x103 PFU, The mice infected dose of WSN or NS1 mutant viruses used in Figure 3C-3G was 104 PFU, This section was modified in the revised manuscript (Page 9, lines 326-327).
3.Was the inoculation dose for mouse experiment using D2I single mutant viruses was greater than that of F9Y experiment? Because the mice inoculated with WSN survived without weight loss for 14 days as shown in Figure 3A, but WSN-inoculated mice all died in Figure 7A.
Thank you for your valuable question.The inoculation dose for mouse experiment using D2I single mutant viruses was 104 PFU, the dose used in Figure 3A was 5x103 PFU.
4.Experimental to examine cytokine mRNA expression was not described in Materials and Methods. The gene expression was shown in n-fold as relative expression level in Figure 7E. The control group is mock mouse (PBS-infected mice)?
Response: Thank you for your valuable advice. The additional description of cytokines mRNA expression in the viruses infected mice was added in Materials and Methods (Page 4, lines 187-191;page 4-5, lines 197-199). The control group is PBS-infected mice. The transcript levels of IL-4 and IL6 in the NS1 F9Y mutant virus infection group were about 6.7- and 3.5-fold higher than those in the WSN infection group. This section was modified in the revised manuscript (Page 14, lines 431-434).
“Results”
5.In Figure 5B, the expression level of TLR7 mRNA at 6 hpi in this heat map seemed to be same between WSN and NS1 F9Y. Both were higher than that of mock. Likewise, the expression levels of TRIM25 and TLR3 at 12 hpi looks same between WSN and NS1 F9Y. Alternatively, OASL expression level was different at 12 hpi. Those could not correspond to the description of results section (lines 349-354). Could you check them?
Response: Thank you for your valuable comment. We have checked the expression level of TLR7 mRNA at 6 hpi , TRIM25 and TLR3 at 12 hpi , the OASL expression level at 12 hpi in this heat map. The expression level of cytokines is the relative expression level of MOCK group as control. The average expression level of TLR7 mRNA at 6 hpi induced by rNS1 F9Y and WSN were 1.85 and 2.18 (p<0.05), levels of TRIM25 mRNA at 12 hpi induced by rNS1 F9Y and WSN were 2.16 and 3.18 (p<0.01), levels of TLR3 mRNA at 12 hpi induced by rNS1 F9Y and WSN were 1.56 and 4.66 (p<0.0001). There was no significant difference in the expression of the OASL mRNA induced by the WSN and rNS1 F9Y viruses due to the large differences between the three biological replicates in the WSN infected group.
6.Could you please add the explanation of bars (rNS1-D2I and WSN) in Figure 6C (like Figure 7E).
Response: Thank you for your valuable suggestion. The explanation bars of (rNS1-D2I and WSN) in Figure 6C have been added in the new manuscript.
7.At line 401, could you show the specific name of mutant viruses (D2I and D2I+T58S), instead of “the mutants”, to distinguish from the description of Figure 3.
Response: Thank you for your valuable advice. The word “the mutants” was rewritten as “rNS1 D2I and rNS1 D2I+T58S” in the revised manuscript. (Page 13, lines 419).
8.At line 401, D21 is typo? D2I may be correct.
Response: Thank you for your valuable comment. The word “D21” was rewritten as “D2I” in the revised manuscript. (Page 13, lines 420).
9.Could you explain about “IntDEM” described at line 427 and Figure 8B?
Response: Thank you for your valuable comment. The word “IntDEM” was rewritten as “IntDEN” in the revised manuscript. The ratio of IntDEN (Integrated density) within nuclear to cytoplasmic indicates nuclear export activity of NS2 protein. The additional description of method has been added to the revised manuscript (Page 5, lines 211-213).
10.In Figure 8B, there were two WSN-NEP-F9Y and no WSN-NEP.
Response: Thank you for your valuable advice. This section was modified in the revised manuscript.
“Discussion”
Dear Editor,
I with my co-authors, appreciate for your letter on our manuscript entitled “D2I and F9Y Mutations in the NS1 Protein of Influenza A Virus Affect Viral Replication Via Regulating Host Innate Immune Responses”. Thank you very much for giving us an opportunity to revise our manuscript. We are also very grateful to the reviewers for their valuable comments by cautious revision, which helps to improve our manuscript. Based on reviewer’s suggestions, we answered each question one by one. We have made careful revision on the original manuscript, all revised portions marked in yellow in the revised manuscript.
Most sincerely, Corresponding author
Jihui Ping
May. 25 2022
Please see our responses as follows:
Reviewer(s)' Comments to Author:
Reviewer4:
Comments to the Author
The manuscript of Yu et al. describes a WSN backbone virus with NS1 F9Y mutation enhanced viral growth in MDCK and A549 cells, and pathogenicity in mice. Moreover, NS1 D2I mutation decreased the virus replication in cells and attenuated pathogenicity in mice. The changes of replication and pathogenicity of these NS1 mutant viruses were likely associated to ability of viral NS1 protein to suppress host innate immune responses such as induction of IFN-beta and other cytokines. I think the authors conducted sufficient experiments and provided persuasive data to support their conclusion. The result part seemed to be a little long but was described well to understand the study. It may need a further explanation at several parts of results (e.g., mouse experiment with D2I mutant and immunofluorescence).
To improve the manuscript, could the authors consider some specific points as following?
“Introduction”
1.Introduction section should be divided to several paragraphs adequately. It is same to Discussion section.
Thank you for your valuable advice. The introduction and discussion section have been divided to several paragraphs adequately.
“Materials and Methods”
2.8 Mouse experiments
2.The inoculation doses of rWSN virus (5x10^3 PFU/50uL) and rNS1 viruses (10^4 PFU/50uL) was different. Did the 2-fold difference have no effect on the viral titer in lungs or weight loss of inoculated mice?
Thank you for your question. The mice infected dose of WSN or NS1 mutant viruses used in Figure 3A and 3B was 5x103 PFU, The mice infected dose of WSN or NS1 mutant viruses used in Figure 3C-3G was 104 PFU, This section was modified in the revised manuscript (Page 9, lines 326-327).
3.Was the inoculation dose for mouse experiment using D2I single mutant viruses was greater than that of F9Y experiment? Because the mice inoculated with WSN survived without weight loss for 14 days as shown in Figure 3A, but WSN-inoculated mice all died in Figure 7A.
Thank you for your valuable question.The inoculation dose for mouse experiment using D2I single mutant viruses was 104 PFU, the dose used in Figure 3A was 5x103 PFU.
4.Experimental to examine cytokine mRNA expression was not described in Materials and Methods. The gene expression was shown in n-fold as relative expression level in Figure 7E. The control group is mock mouse (PBS-infected mice)?
Response: Thank you for your valuable advice. The additional description of cytokines mRNA expression in the viruses infected mice was added in Materials and Methods (Page 4, lines 187-191;page 4-5, lines 197-199). The control group is PBS-infected mice. The transcript levels of IL-4 and IL6 in the NS1 F9Y mutant virus infection group were about 6.7- and 3.5-fold higher than those in the WSN infection group. This section was modified in the revised manuscript (Page 14, lines 431-434).
“Results”
5.In Figure 5B, the expression level of TLR7 mRNA at 6 hpi in this heat map seemed to be same between WSN and NS1 F9Y. Both were higher than that of mock. Likewise, the expression levels of TRIM25 and TLR3 at 12 hpi looks same between WSN and NS1 F9Y. Alternatively, OASL expression level was different at 12 hpi. Those could not correspond to the description of results section (lines 349-354). Could you check them?
Response: Thank you for your valuable comment. We have checked the expression level of TLR7 mRNA at 6 hpi , TRIM25 and TLR3 at 12 hpi , the OASL expression level at 12 hpi in this heat map. The expression level of cytokines is the relative expression level of MOCK group as control. The average expression level of TLR7 mRNA at 6 hpi induced by rNS1 F9Y and WSN were 1.85 and 2.18 (p<0.05), levels of TRIM25 mRNA at 12 hpi induced by rNS1 F9Y and WSN were 2.16 and 3.18 (p<0.01), levels of TLR3 mRNA at 12 hpi induced by rNS1 F9Y and WSN were 1.56 and 4.66 (p<0.0001). There was no significant difference in the expression of the OASL mRNA induced by the WSN and rNS1 F9Y viruses due to the large differences between the three biological replicates in the WSN infected group.
6.Could you please add the explanation of bars (rNS1-D2I and WSN) in Figure 6C (like Figure 7E).
Response: Thank you for your valuable suggestion. The explanation bars of (rNS1-D2I and WSN) in Figure 6C have been added in the new manuscript.
7.At line 401, could you show the specific name of mutant viruses (D2I and D2I+T58S), instead of “the mutants”, to distinguish from the description of Figure 3.
Response: Thank you for your valuable advice. The word “the mutants” was rewritten as “rNS1 D2I and rNS1 D2I+T58S” in the revised manuscript. (Page 13, lines 419).
8.At line 401, D21 is typo? D2I may be correct.
Response: Thank you for your valuable comment. The word “D21” was rewritten as “D2I” in the revised manuscript. (Page 13, lines 420).
9.Could you explain about “IntDEM” described at line 427 and Figure 8B?
Response: Thank you for your valuable comment. The word “IntDEM” was rewritten as “IntDEN” in the revised manuscript. The ratio of IntDEN (Integrated density) within nuclear to cytoplasmic indicates nuclear export activity of NS2 protein. The additional description of method has been added to the revised manuscript (Page 5, lines 211-213).
10.In Figure 8B, there were two WSN-NEP-F9Y and no WSN-NEP.
Response: Thank you for your valuable advice. This section was modified in the revised manuscript.
“Discussion”
11.At line 489, I don’t know which is better, NS2 mutations or NEP mutations.
Response: Thank you for your valuable advice. This section was modified in the revised manuscript (Page 16, lines 510).
11.At line 489, I don’t know which is better, NS2 mutations or NEP mutations.
Response: Thank you for your valuable advice. This section was modified in the revised manuscript (Page 16, lines 510).
